# Assessing the Impact of Image Resolution on Deep Learning for TB Lesion Segmentation on Frontal Chest X-rays

**DOI:** 10.3390/diagnostics13040747

**Published:** 2023-02-16

**Authors:** Sivaramakrishnan Rajaraman, Feng Yang, Ghada Zamzmi, Zhiyun Xue, Sameer Antani

**Affiliations:** Computational Health Research Branch, National Library of Medicine, National Institutes of Health, Bethesda, MD 20894, USA

**Keywords:** aspect ratio, chest X-ray, deep learning, image resolution, segmentation, tuberculosis, test-time augmentation, threshold selection

## Abstract

Deep learning (DL) models are state-of-the-art in segmenting anatomical and disease regions of interest (ROIs) in medical images. Particularly, a large number of DL-based techniques have been reported using chest X-rays (CXRs). However, these models are reportedly trained on reduced image resolutions for reasons related to the lack of computational resources. Literature is sparse in discussing the optimal image resolution to train these models for segmenting the tuberculosis (TB)-consistent lesions in CXRs. In this study, we investigated the performance variations with an Inception-V3 UNet model using various image resolutions with/without lung ROI cropping and aspect ratio adjustments and identified the optimal image resolution through extensive empirical evaluations to improve TB-consistent lesion segmentation performance. We used the Shenzhen CXR dataset for the study, which includes 326 normal patients and 336 TB patients. We proposed a combinatorial approach consisting of storing model snapshots, optimizing segmentation threshold and test-time augmentation (TTA), and averaging the snapshot predictions, to further improve performance with the optimal resolution. Our experimental results demonstrate that higher image resolutions are not always necessary; however, identifying the optimal image resolution is critical to achieving superior performance.

## 1. Introduction

*Mycobacterium tuberculosis* (MTB) is the cause of pulmonary tuberculosis (TB) [1]; however, it can also affect other body organs including the brain, spine, and kidneys. TB infection can be categorized into latent and active types. Latent TB refers to cases where the MTB remains inactive and causes no symptoms. Active TB is contagious and can spread to others. The Centers for Disease Control and Prevention recommends people having an increased risk of acquiring TB infection including those with HIV/AIDS, using intravenous drugs, and from countries with a high prevalence, be screened for the disease [2]. Chest X-ray (CXR) is the most commonly used radiographic technique to screen for cardiopulmonary abnormalities, particularly TB [3]. Some of the TB-consistent abnormal manifestations in the lungs include apical thickening; calcified, non-calcified, and clustered nodules; infiltrates; cavities; linear densities; adenopathy; miliary patterns; and retraction, among others [1]. These manifestations can be observed anywhere in the lungs and may vary in size, shape, and density.

While CXRs are widely adopted for TB infection screening, human expertise is scarce [4], particularly in low and middle-resourced regions, for reading the CXRs. The development of machine learning-based (ML) artificial intelligence (AI) tools could aid in the screening through automated segmentation of disease-consistent regions of interest (ROIs) in the images.

## 2. Related Literature and Contributions of the Study

Currently, deep learning (DL) models, a subset of ML algorithms, are observed to perform on par with human experts in segmenting body organs such as the lungs, heart, clavicles [5,6], and other cardiopulmonary disease manifestations including brain tumor [7,8,9], COVID-19 [10], pneumonia [11], and TB [12] in CXRs. These CXRs are made publicly available at high resolutions. Digital CXRs typically have a full resolution of approximately 2000 × 2500 pixels [13]; however, these may vary based on the sensor matrix. For instance, the CXRs in the Shenzhen CXR data collection [14] have an average resolution of 2644-pixel width × 2799-pixel height. However, a majority of current segmentation studies [15,16,17] are conducted using CXRs that are down-sampled to 224 × 224 pixel resolution due to GPU constraints. An extensive reduction in image resolution may eliminate subtle or weakly-expressed disease-relevant information. This important information may be hidden in small details, such as the surface and contour of the lesion, and other patterns in findings. As the details preserved in the visual information can drastically vary with the changes in image resolution and the type of subsampling method used, we believe the choice of image resolution should not depend on the computational hardware availability, but rather on the characteristics of the data.

Our review of the literature revealed the importance of image resolution and its impact on performance. For example, the authors in [18] found that changes in endoscopy image resolution impact classification performance. Another study [19] reported an improved disease classification performance at lower CXR image resolutions. The authors observed that the overfitting issues were resolved at lower input image resolutions. Our review of the literature also revealed that identifying the optimal image resolution for the task under study remains an open avenue for research. Until the writing of this manuscript, we have not found any study that discussed the impact of image resolution on a CXR-based segmentation task, particularly for segmenting TB-consistent lesions. To close this gap in the literature, this work aims to study the impact of training a model on varying image resolutions with/without lung ROI cropping and aspect ratio adjustments to find the optimal resolution that improves fine-grained TB-consistent lesion segmentation. Further, this work proposes to improve performance at the optimal resolution through a combinatorial approach consisting of storing model snapshots, optimizing the test-time augmentation (TTA) methods, optimizing the segmentation threshold, and averaging the predictions of the model snapshots.

Section 3 discusses the materials and methods. Section 4 elaborates on the results, and Section 5 discusses and concludes this study.

## 3. Materials and Methods

### 3.1. Data Characteristics

This study uses the Shenzhen CXR dataset [14] collected at the Shenzhen No. 3 hospital, in Shenzhen, China. The CXRs were de-identified at the source and are made available by the National Library of Medicine (NLM). The dataset contains 336 CXRs collected from microbiologically confirmed TB cases and 326 CXRs showing normal lungs. Table 1 shows the dataset characteristics.

The CXRs manifesting TB were annotated by two radiologists from the Chinese University of Hong Kong. The labeling was initially conducted by a junior radiologist, and then the labels were all checked by a senior radiologist, with a consensus reached for all cases. The annotations were stored as both binarized masks as well as pixel boundaries stored in JSON format [1]. The authors of [20] manually segmented the lung regions and made them available as lung masks. These masks are available for 287 CXRs manifesting TB-consistent abnormalities and 279 CXRs showing normal lungs. We used these 287 TB CXRs out of 336 TB CXRs that have both lung masks and TB lesion-consistent masks. Figure 1 shows the following: (a) The binarized TB masks of men and women were resized to 256 × 256 to maintain uniformity in scale. Then, the masks were averaged, normalized to the range [0, 1], and displayed using the “jet” colormap. (b) Pie chart showing the proportion and distribution of TB in men and women. (c) Age-wise distribution of the normal and TB-infected population of men and women.

These 287 CXRs were further divided at the patient level into 70% for training (*n* = 201), 10% for validation (*n* = 29), and 20% for hold-out testing (*n* = 57). The masks were thresholded and binarized to separate the foreground lung/TB-lesion pixels from the background pixels.

### 3.2. Model Architecture

We used the Inception-V3 UNet model architecture that we have previously demonstrated [12] to deliver superior TB-consistent lesion segmentation performance. The Inception-V3-based encoder [21] was initialized with ImageNet weights. The model was trained for 128 epochs at various image resolutions and is discussed in Section 3.3. We used an Adam optimizer with an initial learning rate of 1 × 10^−3^ to minimize the boundary-uncertainty augmented focal Tversky loss [8]. The learning rate was reduced if the validation loss ceased to improve after 5 epochs. This is called the *patience* parameter; its value was chosen from pilot evaluations. We stored the model weights whenever the validation loss decreased. The best-performing model with the validation data was used to predict the test data. The models were trained using Keras with Tensorflow backend (*ver. 2.7*) using a single NVIDIA GTX 1080 Ti GPU and CUDA dependencies.

### 3.3. Image Resolution

We empirically identified the optimal image resolution at which the Inception-V3 UNet model delivered superior performance toward the TB-consistent lesion segmentation task. The model was trained using various image/mask resolutions, viz., 32 × 32, 64 × 64, 128 × 128, 256 × 256, 512 × 512, 768 × 768, and 1024 × 1024. We used a batch size of 128, 64, 32, 16, 8, 4, and 2, respectively. We used bicubic interpolation to down-sample the 287 CXR images and their associated TB masks to the aforementioned resolutions, as shown in Figure 2. As expected, the visual details improved with increasing resolution. 

We evaluated the model performance under the following conditions:(i)The 287 CXRs and their associated TB masks were directly down-sampled using bi-cubic interpolation to the aforementioned resolutions. The OpenCV package (*ver. 4.5.4*) was used in this regard.(ii)The lung masks were overlaid on the CXRs and their associated TB masks to delineate the lung boundaries. The lung ROI was cropped to the size of a bounding box and also down-sampled to the aforementioned resolutions.(iii)Based on performance, the data from step (i) or step (ii) was corrected for aspect ratio, the details are discussed in Section 3.4. The corrected aspect-ratio CXRs/masks were further down-sampled to the aforementioned resolutions.

### 3.4. Aspect Ratio Correction

The aspect ratio is defined as the ratio of width to height [22]. To find the aspect ratio, the mean and standard deviation of the widths and heights of the CXRs manifesting TB-consistent abnormalities were computed. For the original CXRs, we observed that the width and height are 2644 ± 253 pixels and 2799 ± 206 pixels, respectively. For the lung-cropped CXRs, we observed that the width and height are 1929 ± 151 pixels and 1999 ± 231 pixels, respectively. For the original CXRs, the computed aspect ratio is 0.945. For the lung-cropped CXRs, the computed aspect ratio is 0.965. We maintained the larger dimension (i.e., height) as constant at various image resolutions and modified the smaller dimension (i.e., width) to adjust the aspect ratio. We constrained the width and height of the images/masks to be divisible by 32 to be compatible with the UNet architecture [23]. For this, we padded the images such that the width was to the nearest lower value that is divisible by 32.

### 3.5. Performance Evaluation

The trained models were evaluated using (i) pixel-wise metrics [24], consisting of the intersection of union (IoU) and Dice score, and (ii) image-wise metrics, consisting of structural similarity index measure (SSIM) [25,26] and signal-to-reconstruction error ratio (SRE) [27]. While IoU and Dice are the most commonly used metrics to evaluate segmentation performance, a study of the literature [28] reveals that pixel-wise metrics ignore the dependencies among the neighboring pixels. The authors of [29] minimized a loss function derived from SSIM to segment ROIs in the Cityscapes and PASCAL VOC 2012 datasets. It was found that the masks predicted by the model that was trained to minimize the SSIM loss were more structurally similar to the ground truth masks compared to the model trained using the conventional cross-entropy loss. Motivated by this study, we used the SSIM metric to evaluate the structural similarity between the ground truth and predicted TB masks.

The SSIM of a pair of images (a, b) is given by a multiplicative combination of the structure (*s*), contrast (*c*), and luminance (*l*) factors, as given in Equation (1):(1)$$SSIM\;\left(a,\;b\right)={ [l\left(a,\;b\right)]}^{\alpha }.{ [c\left(a,\;b\right)]}^{\beta }.{ [s\left(a,b\right)]}^{\;\gamma }$$

The luminance (*l*) is measured by averaging over all the image pixel values. It is given by Equation (2). The luminance comparison between a pair of images (a, b) is given by a function of ${µ}_{a}$ and ${µ}_{b}$, as shown in Equation (3):(2)$${µ}_{a}=\frac{1}{N}\overset{N}{\underset{i=1}{\sum }}a_{i}$$
(3)$$l\left(a,b\right)=\frac{2{µ}_{a}{µ}_{b}+C_{1}}{{µ}_{a}^{2}+{µ}_{b}^{2}+C_{1}}$$

The contrast (*c*) is measured by taking the square root of the variance of all the image pixel values. The comparison of contrast between two images (a, b) is given by Equation (4) and Equation (5):(4)$$\sigma_{a}=(\frac{1}{N-1}\overset{N}{\underset{i=1}{\sum }}{\left(a_{i}-{µ}_{a}\right)}^{2}{)}^{1/2}$$
(5)$$c\left(a,b\right)=\frac{2\sigma_{a}\sigma_{b}+C_{2}}{\sigma_{a}^{2}+\sigma_{b}^{2}+C_{2}}$$

The structural (*s*) comparison is given by dividing the input by its standard deviation, as shown in Equations (6) and (7):(6)$$s\left(a,b\right)=\frac{\sigma_{ab}+C_{3}}{\sigma_{a}\sigma_{b}+C_{3}}$$
(7)$$\sigma_{ab}=\frac{1}{N-1}\;\overset{N}{\underset{i=1}{\sum }}\left((a_{i}-{µ}_{a}\right)\left((b_{i}-{µ}_{b}\right)$$

The constants $C_{1,\;}C_{2},\;C_{3}$ ensure numerical stability when the denominator becomes 0. The value of IoU, Dice, and SSIM range from [0, 1].

We visualized the SSIM quality map (using “jet” colormap) to interpret the quality of the predicted masks. The quality map is identical in size to the corresponding scaled version of the image. Small values of SSIM appear as dark blue activations, denoting regions of poor similarity to the ground truth. Large values of SSIM appear as dark red activations, denoting regions of high similarity.

The authors of [27] proposed a metric called signal-to-reconstruction error ratio (SRE) that measures the error relative to the mean image intensity. The authors discussed that the SRE metric is robust to brightness changes while measuring the similarity between the predicted image and ground truth. The SRE metric is measured in decibels (dB) and is given by Equation (8):(8)$$SRE=10\;log_{10}\left(\frac{{µ}_{a}^{2}}{\frac{{\left|\left|\hat{a}-a\right|\right|}^{2}}{n}}\right)$$
where, ${µ}_{a}$ denotes the average value of the image a and n denotes the number of pixels in image a.

### 3.6. Optimizing the Segmentation Threshold

Studies in the literature [15,30,31] used a threshold of 0.5 by default in segmentation tasks. However, the process of selecting the segmentation threshold should be driven by the data under study. An arbitrary threshold of 0.5 is not guaranteed to be optimal, particularly considering imbalanced data, as in our case, where the number of foreground TB-consistent lesion pixels is considerably smaller compared to the background pixels. It is therefore important to perform a threshold tuning, in which we iterate among different segmentation threshold values in the range of [0, 1] and find the optimal threshold that would maximize performance. In our case, we generated 200 equally spaced samples in the closed interval [0, 1] and used a looping mechanism to find the optimal segmentation threshold that maximized the IoU metric for the validation data. This threshold was used to binarize the predicted masks using the test data and the performance was measured in terms of the evaluation metrics discussed in Section 3.5.

### 3.7. Storing Model Snapshots at the Optimal Resolution

After we empirically identified the optimal resolution, we further improved performance at this resolution as follows: (i) we adopted a method called “snapshot ensembling” [32], which involves using an aggressive cyclic learning rate to train and store diversified model snapshots (i.e., the model weights) during a single training run; (ii) we initialized the training process with a high learning rate of 1 × 10^−2^, defined the number of training epochs as 320, and the number of training cycles as 8 so that each training cycle is composed of 40 epochs; (iii) the learning rate was rapidly decreased to the minimum value of 1 × 10^−8^ at the end of each training cycle before being drastically increased during the next cycle. This acts similar to a simulated restart, resulting in using good weights as the initialization for the subsequent cycle, thereby allowing the model to converge to different local optima; (iv) the weights at the bottom of each cycle are stored as snapshots (with 8 training cycles, we stored 8 model snapshots); (v) we evaluated the validation performance of each of these snapshots at their optimal segmentation threshold identified as discussed in Section 3.6. This threshold was further used to binarize the predicted test data and the performance was measured.

### 3.8. Test-Time Augmentation (TTA)

Test-time augmentation (TTA) refers to the process of augmenting the test set [33]. That is, the trained model predicts the original and transformed versions of the test set, and the predictions are aggregated to produce the final result. One advantage of performing TTA is that no changes are required to be made to the trained model. TTA ensures diversification and helps the model with improved chances of better capturing the target shape, thereby improving model performance and eliminating overconfident predictions. However, these studies [33,34,35] are observed to perform multiple random image augmentations without identifying the optimal augmentation method(s) that would help improve performance. A possible negative effect of destroying/degrading visual information with non-optimal augmentation(s) might outweigh the benefit of augmentation while also resulting in increased computational load.

After storing the model snapshots as discussed in Section 3.7, we performed TTA with the validation data using each model snapshot. In addition to the original input, we used the augmentation methods consisting of horizontal flipping, pixel-wise width, height shifting (−5, 5), and rotation in degrees (−5, 5) individually and in combination, as shown in Table 2.

For each TTA combination shown in Table 2, an aggregation function takes the set of predictions and averages them to produce the final prediction. We identified the optimal segmentation threshold that maximized the IoU for each model snapshot and every TTA combination. With the identified optimal TTA augmentation combination and the segmentation threshold, we augmented the test data, recorded the predictions, binarized them, and evaluated performance. This process is illustrated in Figure 3. We further constructed an ensemble of the top-K (K = 2, 3, …, 6) by averaging their predictions. We call this *snapshot averaging*. The pseudocode explaining our proposal is shown in Figure 4.

### 3.9. Statistical Analysis

We measured the 95% binomial Clopper–Pearson confidence intervals (CIs) for the IoU metric obtained at various stages of our empirical analyses.

## 4. Results

Table 3 shows the performance achieved through training the Inception-V3 UNet model using the CXRs/TB masks of varying image resolutions, viz., 32 × 32, 64 × 64, 128 × 128, 256 × 256, 512 × 512, 768 × 768, and 1024 × 1024. Figure 5 shows the sample predictions at these resolutions. The performances are reported for each image resolution at its optimal segmentation threshold. The term *O* and *CR* denote the original and lung-cropped CXRs/masks, respectively. We observed poor performance at 32 × 32 resolution with both original and lung-cropped data.

The performance kept improving until 256 × 256-pixel resolution where the model achieved the best IoU of 0.4859 (95% CI: (0.3561, 0.6157)) and superior values for Dice, SSIM, and SRE metrics. The performance then kept decreasing from 256 × 256 to 1024 × 1024 resolution. The performance achieved with the lung-cropped data is superior compared to the original counterparts at all resolutions. These observations highlighted that 256 × 256 is the optimal resolution and using lung-cropped CXRs/masks gave a superior performance.

Figure 6 shows the SSIM quality maps achieved by the Inception-V3 UNet model for a sample test CXR at varying image resolutions. The quality maps are identical in size to the corresponding scaled version of the images/masks. We observed high activations, shown as red pixels, in regions where the predicted masks were highly similar to the ground truth masks. Blue pixel activations denote regions of poor similarity. We observed the following: (i) The predicted masks exhibited poor similarity to the ground truth masks along the mask edges for all image resolutions. (ii) The SSIM value obtained with the lung-cropped data was superior compared to the original counterparts.

Table 4 shows the performance achieved by the Inception-V3 UNet model with aspect-ratio corrected (*AR-CR*) lung-cropped CXRs/masks for varying image resolutions. We observed no improvement in performance with aspect-ratio corrected data at any given image resolution compared to the results reported in Table 3.

To improve performance at the optimal image resolution, i.e., 256 × 256, we stored the model snapshots, as discussed in Section 3.7, and performed TTA augmentation for each recorded snapshot, as discussed in Section 3.8. Table 5 shows the optimal TTA combinations that delivered superior performance for each model snapshot at its optimal segmentation threshold identified from the validation data.

The terms S1, S2, S3, S4, S5, S6, S7, and S8 denote the 1^st^, 2^nd^, 3^rd^, 4^th^, 5^th^, 6^th^, 7^th^, and 8^th^ model snapshot, respectively. The TTA combination that aggregates (averages) the predictions of the original test data with those obtained from other augmentations consisting of horizontal flipping, width shifting, height shifting, and rotation, delivered superior performance for the S3, S4, S5, S7, and S8 model snapshots. The aggregation of the original predictions with height-shifting augmentation delivered superior performance for the S2 snapshot. The S6 snapshot delivered superior performance while aggregating the original predictions with those obtained from the width and height-shifted images. Aggregating the predictions of the original test data with those augmented by width, height shifting, and rotation, delivered superior test performance while using the S1 model snapshot. The first row of Table 6 shows the performance achieved by the model trained with the 256 × 256 lung-cropped CXRs/masks, denoted as *CR-baseline* (from Table 3).

Rows 2–9 denote the performance achieved by the model snapshots S1–S8. Rows 10–17 show the performances achieved by the model snapshots at their optimal TTA combination (Table 5). We observed that TTA improved segmentation performance for the recorded model snapshot in terms of all metrics compared to the model snapshots without TTA and the “CR baseline”.

We ranked the model snapshots S1–S8 in terms of their IoU. We observed the S2 snapshot delivered the best IoU, followed by S3, S5, S7, S6, and S4 model snapshots. We constructed an ensemble of the top-K snapshots (K = 2, 3, …, 6), as discussed in Section 3.8, by averaging their predictions obtained using their optimal TTA combination. Rows 18–22 show the performances achieved by the ensemble of the top-2, top-3, top-4, top-5, and top-6 model snapshots, respectively. We observed that the snapshot averaging ensemble constructed using the top-4 and top-5 model snapshots delivered superior performance in terms of the IoU and Dice metrics while the top-5 snapshot ensemble delivered superior values also in terms of the SSIM and SRE metrics. The segmentation performance improved in terms of all evaluation metrics at the optimal 256 × 256 resolution by constructing an averaging ensemble of the top-5 model snapshots compared to the *CR-baseline*.

Figure 7 shows the predictions achieved by the baseline (i.e., the Inception-V3 UNet model trained with the lung-cropped CXRs/masks at the 256 × 256 resolution), and snapshot averaging of the top-5 model snapshots with TTA for a couple of CXRs from the test set. In the first row, we could observe that snapshot averaging removed the false positives (predictions shown with blue contours). In the second row, we could observe that the predicted masks were increasingly similar to the ground truth masks (shown with red contours), compared to the baseline.

Figure 8 shows the SSIM quality maps achieved with the baseline and snapshot averaging for a couple of CXR instances from the test set.

## 5. Discussion and Conclusions

We observed that the segmentation performance improved with increasing image resolution from 32 × 32 up to 256 × 256. The performance achieved with the lung-cropped CXRs/TB-lesion masks was superior compared to their original counterparts. These findings are consistent with [31,36,37,38], in which lung cropping was reported to improve performance in medical image segmentation and classification tasks. We observed that increasing the resolution beyond 256 × 256 decreased segmentation performance. This can be attributed to the fact that (i) increasing resolution also increased the feature space to be learned by the models, and (ii) increased parameter count might have led to model overfitting to the training data because of limited data availability.

We observed that the SSIM value decreased with decreasing resolution. The possible reasons for this reduction are as follows: the SSIM index is based on three components, the luminance component, which compares the average pixel intensity of the two images, the contrast component, which compares the standard deviation of the pixel intensities, and the structural component, which compares the similarity of patterns in the two images. When the resolution of an image is decreased, the number of pixels in the image is reduced, which can lead to a loss of detail in the image. This loss of detail can result in lower values for the luminance and contrast components, which in turn can lead to a lower overall SSIM score. In addition, the structural component of the SSIM index compares the similarity of patterns in the two images using a windowed function, which is sensitive to the resolution of the image. When the resolution is reduced, the window function captures less information and thus, the structural component becomes less effective in capturing the similarities between the two images. However, the SSIM metrics achieved with the lung-cropped images were superior to the original images, and the performance further improved with snapshot averaging.

We did not observe a considerable performance improvement with aspect ratio corrections. We were constrained by the UNet architecture [23], which requires that the length and width of the images/masks should be divisible by 32. This limitation did not allow us to make precise aspect ratio corrections. However, the study of literature [22] revealed that DL models trained on medical images are robust to changes in the aspect ratio. Abnormalities manifesting TB do not have a precise shape and they exhibit a high degree of variabilities such as nodules, effusions, infiltrations, cavitations, miliary patterns, and consolidations, among others. These manifestations would appear with their inherent characteristics that provide diversified features to learn for a segmentation model.

We identified the optimal image resolution and further improved performance at that resolution through a combinatorial approach consisting of storing model snapshots, optimizing the TTA and segmentation threshold, and averaging the snapshot predictions. These findings are consistent with the literature in which storing model snapshots and performing TTA considerably improved performance in natural and medical computer vision tasks [33,39,40,41,42]. We further emphasize that identifying the optimal TTA method(s) is indispensable to achieve superior performance compared to randomly augmenting the test data. We underscore the importance of using the optimal segmentation threshold compared to the conventional threshold of 0.5, as widely discussed in the literature [43,44].

Another limitation is that our experiments and conclusions are based on the Shenzhen CXR dataset where we observed that segmenting TB-consistent lesions using an UNet model trained on lung-cropped CXRs/masks delivers optimal performance at the 256 × 256 image resolution. These observations could vary across the datasets. We, therefore, emphasize that the characteristics of the data under study, the model performances at varying image resolutions with/without ROI cropping, and aspect ratio adjustments should be discussed in all studies.

Due to GPU constraints, we were not able to train high-resolution models at larger batch sizes. However, with the advent of high-performance computing, this can be made feasible. High-resolution datasets might require newer model architecture and hardware advancements. Nevertheless, although the full potential of high-resolution datasets is not explored yet, it is indispensable to collect data at the highest resolution possible. Additionally, irrespective of the image resolution, adding more experts to the annotation process may reduce the variation in the ground truth, which we believe may improve segmentation performance.

## Figures and Tables

**Figure 1 diagnostics-13-00747-f001:**
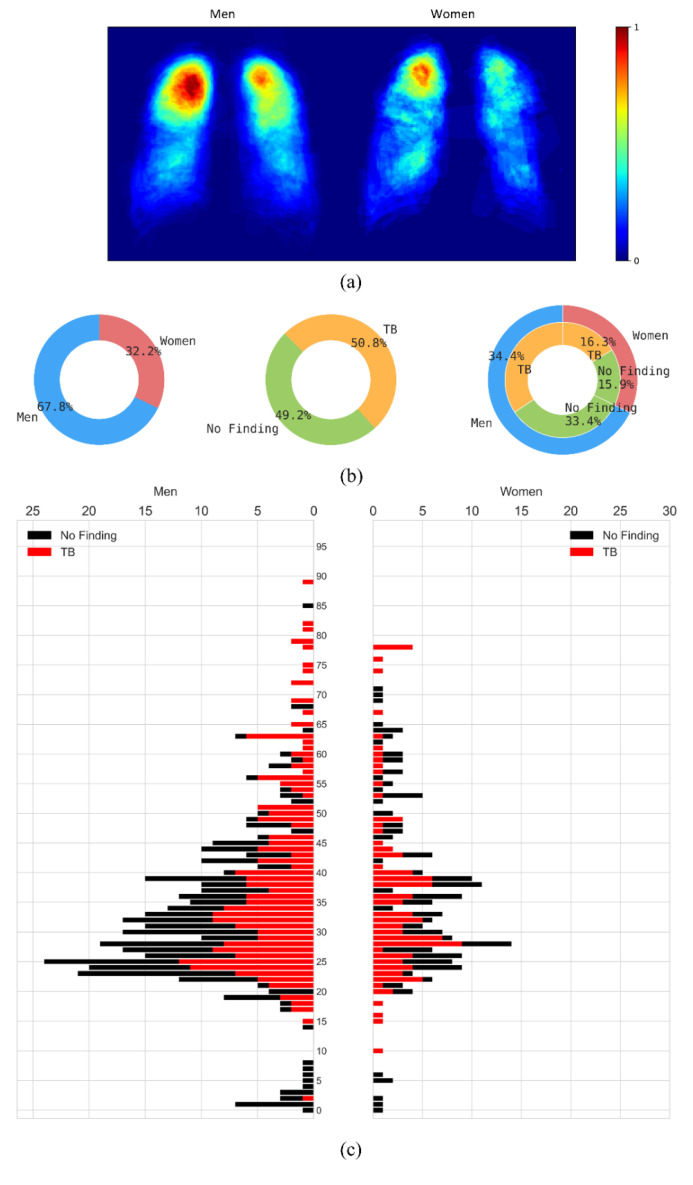
Data characteristics are shown as a proportion of men and women in the Shenzhen CXR collection. (**a**) Heatmaps showing regions of TB infestation in men and women. (**b**) Pie chart showing the proportion and distribution of TB in men and women, and (**c**) Age-wise distribution of the normal and TB-infected population in men and women.

**Figure 2 diagnostics-13-00747-f002:**
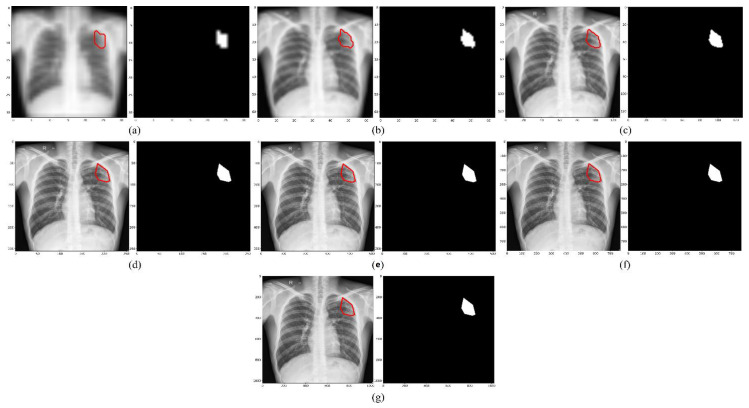
CXRs and their corresponding TB-consistent lesion masks at various image resolutions. (**a**) 32 × 32; (**b**) 64 × 64; (**c**) 128 × 128; (**d**) 256 × 256; (**e**) 512 × 512; (**f**) 768 × 768; and (**g**) 1024 × 1024. All images and masks are rescaled to 256 × 256 to compare quality. The red contours indicate ground truth annotations.

**Figure 3 diagnostics-13-00747-f003:**
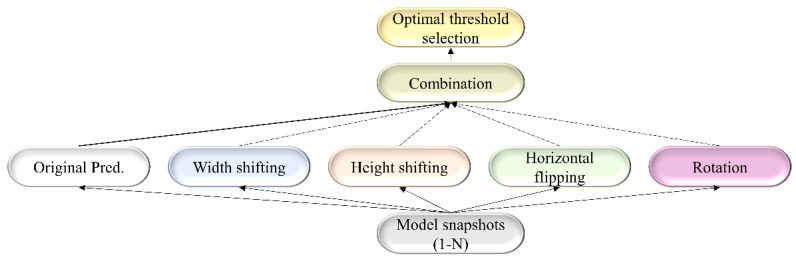
A combinatorial workflow showing the storage of model snapshots and identifying the optimal TTA combination at the optimal segmentation threshold for each snapshot. The term “Original pred.” refers to the model predicting the original, non-augmented data.

**Figure 4 diagnostics-13-00747-f004:**
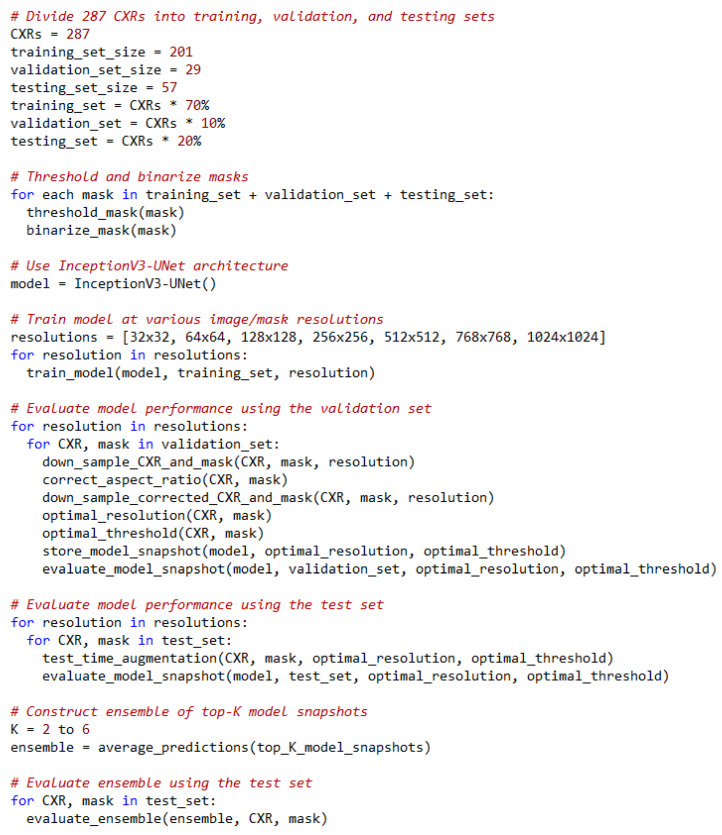
Pseudocode of our proposal.

**Figure 5 diagnostics-13-00747-f005:**
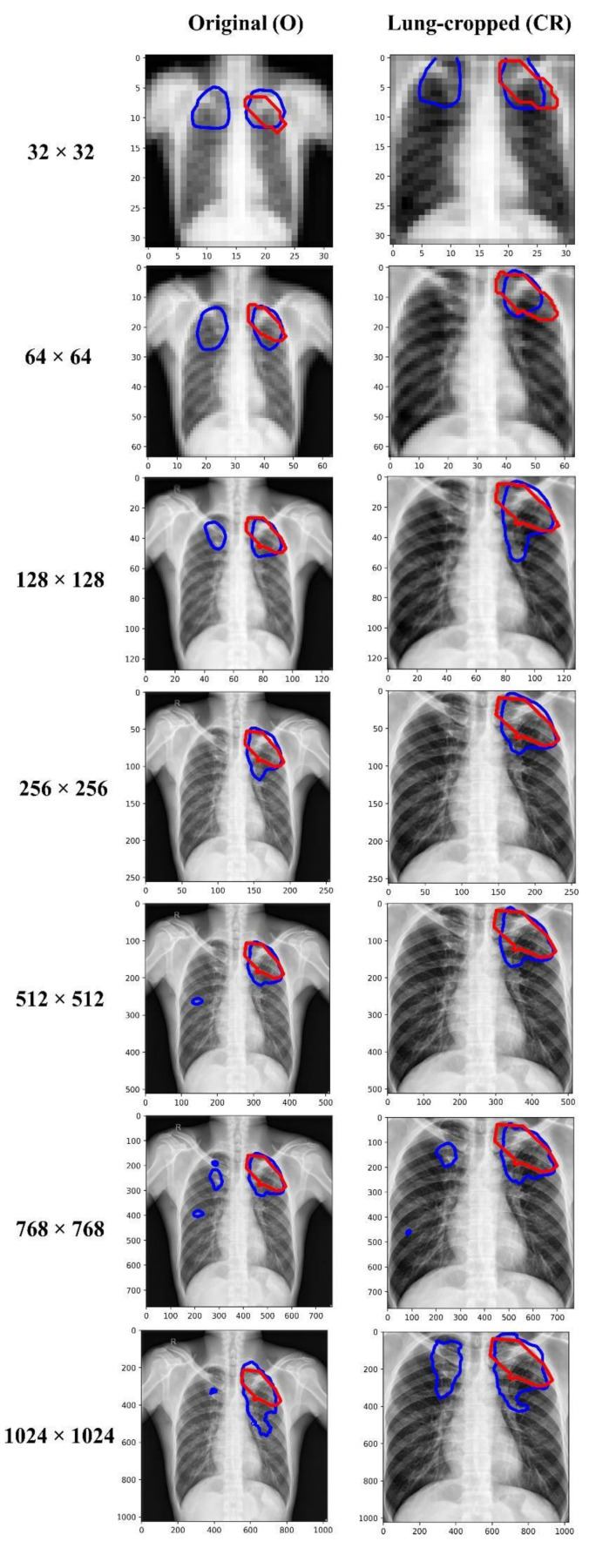
Visualizing and comparing the segmentation predictions of the Inception-V3 UNet model trained at various image resolutions, using a sample original, and its corresponding lung-cropped CXR/mask from the test set. The red and blue contours denote ground truth and predictions, respectively.

**Figure 6 diagnostics-13-00747-f006:**
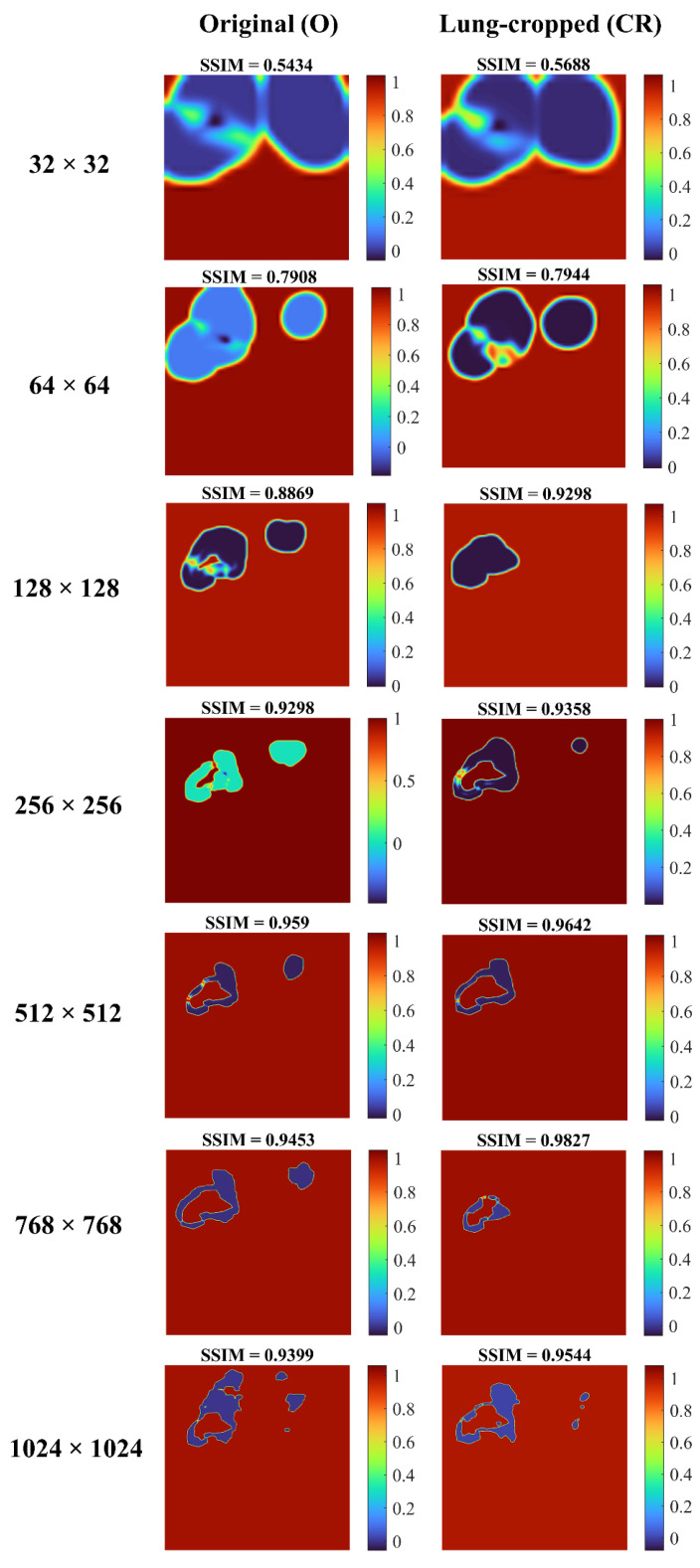
SSIM quality maps are shown for the predictions achieved by the Inception-V3 UNet model trained on various CXR/mask resolutions using a sample original and its corresponding lung-cropped data from the test set.

**Figure 7 diagnostics-13-00747-f007:**
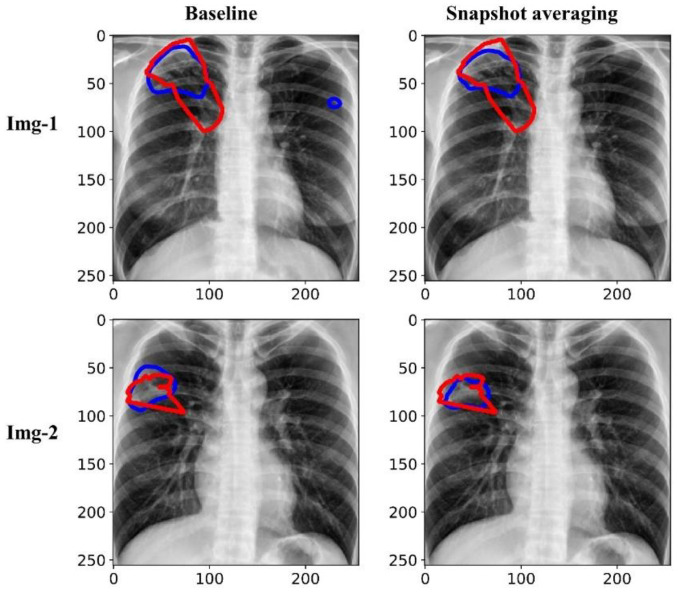
Visualizing and comparing the segmentation predictions of the baseline (i.e., Inception-V3 UNet model trained with lung-cropped CXRs/masks at the 256 × 256 resolution), and the snapshot averaging of the top-5 model snapshots. The red and blue contours denote ground truth and predictions, respectively.

**Figure 8 diagnostics-13-00747-f008:**
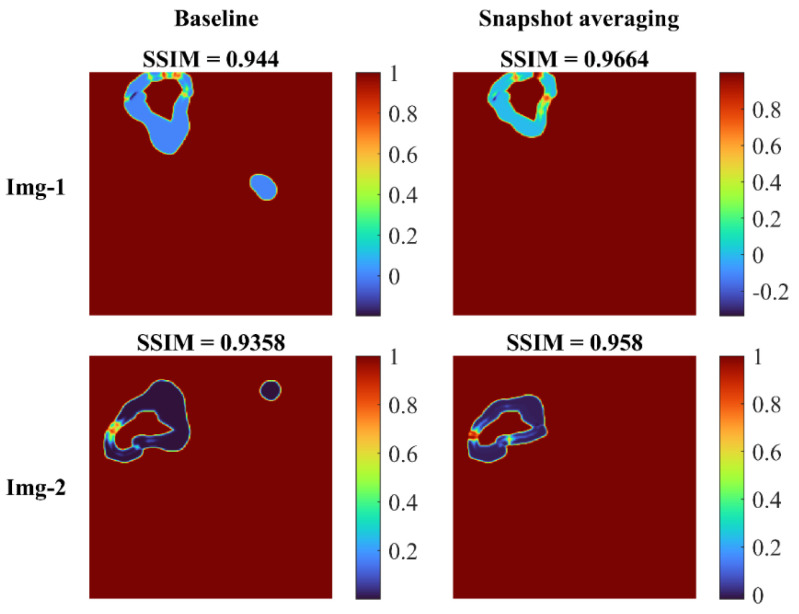
SSIM quality maps shown for the predictions achieved for a couple of test CXRs, by the baseline (Inception-V3 UNet model trained with lung-cropped CXRs/masks at the 256 × 256 resolution), and the snapshot averaging of the top-5 model snapshots with their optimal TTA combination. We observed higher values for the SSIM using the snapshot averaged predictions compared to the baseline, signifying that the predicted masks were increasingly similar to the ground truth masks. Snapshot averaging removed the false positives, and demonstrated improved prediction similarity to the ground truth, with a higher SSIM value, compared to the baseline.

**Table 1 diagnostics-13-00747-t001:** Dataset characteristics. The age of the population of men and women, image width, and image height are given in terms of mean ± standard deviation.

# TB CXRs	# Men	# Women	Age of Men (in Years)	Age of Women (in Years)	# Lung Masks	# TB Masks	Image Width (in Pixels)	Image Height (in Pixels)
336	228	108	38.29 ± 15.12	36.5 ± 14.75	287	336	2644 ± 253	2799 ± 206

# denotes the number of images.

**Table 2 diagnostics-13-00747-t002:** TTA combinations.

Method	TTA Combinations
M1	Original + horizontal flipping
M2	Original + width shifting
M3	Original + height shifting
M4	Original + width shifting + height shifting
M5	Original + horizontal flipping + width shifting + height shifting
M6	Original + rotation
M7	Original + width shifting + height shifting + rotation
M8	Original + horizontal flipping + width shifting + height shifting + rotation

**Table 3 diagnostics-13-00747-t003:** Performance achieved by the Inception-V3-UNet model with original and lung-cropped CXRs and TB-lesion-consistent masks. The term SRE, O, CR, and Opt. T denotes signal-to-reconstruction error ratio, original CXRs and TB-lesion-consistent masks, lung-ROI-cropped CXRs and TB-lesion-consistent masks, and the optimal segmentation threshold. Values in parenthesis denote the 95% CIs as the Exact measure of the Clopper–Pearson interval for the IoU metric. The bold numerical values denote superior performance for the respective columns.

Resolution	IoU	Dice	SSIM	SRE	Opt. T
32 × 32 (O)	0.2183 (0.1110, 0.3256)	0.3583	0.3725	19.9014	0.9548
32 × 32 (CR)	0.2934 (0.1751, 0.4117)	0.4537	0.4414	22.5763	0.6332
64 × 64 (O)	0.3105 (0.1903, 0.4307)	0.4739	0.5548	20.5444	0.3719
64 × 64 (CR)	0.3789 (0.2529, 0.5049)	0.5496	0.5584	24.4192	0.1005
128 × 128 (O)	0.4298 (0.3012, 0.5584)	0.6012	0.6694	23.1622	0.2663
128 × 128 (CR)	0.4652 (0.3357, 0.5947)	0.6350	0.7028	30.1203	0.0704
256 × 256 (O)	0.4567 (0.3273, 0.5861)	0.6271	0.7456	25.3184	0.9900
256 × 256 (CR)	**0.4859 (0.3561, 0.6157)**	**0.6540**	0.7720	29.1329	0.9950
512 × 512 (O)	0.4435 (0.3145, 0.5725)	0.6144	0.8327	27.6090	0.9799
512 × 512 (CR)	0.4799 (0.3502, 0.6096)	0.6485	0.8788	31.7887	0.9950
768 × 768 (O)	0.4428 (0.3138, 0.5718)	0.6138	0.8683	29.3264	0.9899
768 × 768 (CR)	0.4512 (0.3220, 0.5804)	0.6219	0.9073	33.3214	0.9899
1024 × 1024 (O)	0.2746 (0.1587, 0.3905)	0.4309	0.8545	28.4218	0.9796
1024 × 1024 (CR)	0.3387 (0.2158, 0.4616)	0.5060	0.8796	33.3320	0.9950

**Table 4 diagnostics-13-00747-t004:** Performance achieved by the Inception-V3-UNet model with the aspect-ratio corrected lung-cropped (*AR-CR*) CXRs and TB-lesion-consistent masks. The image resolutions are given in terms of height × width.

Resolution (AR-CR)	IoU	Dice	SSIM	SRE	Opt. T
64 × 32	0.1583 (0.0635, 0.2531)	0.2734	0.1884	21.5695	0.9950
128 × 96	0.3474 (0.2237, 0.4711)	0.5157	0.5175	25.2175	0.9950
256 × 224	0.4447 (0.3156, 0.5738)	0.6151	0.7336	28.8964	0.9698
512 × 480	0.4815 (0.3517, 0.6113)	0.6500	0.8333	31.7451	0.9796
768 × 736	0.4200 (0.2918, 0.5482)	0.5916	0.8544	32.8540	0.9796
1024 × 960	0.3259 (0.2042, 0.4476)	0.4915	0.8710	33.6026	0.0204

**Table 5 diagnostics-13-00747-t005:** Optimal test-time augmentation combination for each model snapshot.

Snapshot	Opt. TTA Combination
S1	Original+ width shifting + height shifting + rotation
S2	Original + height shifting
S3	Original+ horizontal flipping + width shifting + height shifting + rotation
S4	Original+ horizontal flipping + width shifting + height shifting + rotation
S5	Original+ horizontal flipping + width shifting + height shifting + rotation
S6	Original+ width shifting + height shifting
S7	Original+ horizontal flipping + width shifting + height shifting + rotation
S8	Original+ horizontal flipping + width shifting + height shifting + rotation

**Table 6 diagnostics-13-00747-t006:** Performance achieved by each model snapshot before and after applying the optimal TTA and averaging the snapshots after TTA. Bold numerical values denote superior performance in respective columns.

Model	IoU	Dice	SSIM	SRE	Opt. T
256 × 256 (CR-Baseline)	0.4859 (0.3561, 0.6157)	0.6540	0.7720	29.1329	0.9950
S1	0.4880 (0.3582, 0.6178)	0.6559	0.7676	29.0406	0.9950
S2	0.5090 (0.3792, 0.6388)	0.6746	0.7937	29.4457	0.9698
S3	0.5024 (0.3725, 0.6323)	0.6688	0.7900	29.4709	0.9749
S4	0.4935 (0.3637, 0.6233)	0.6609	0.7872	29.4803	0.9296
S5	0.4974 (0.3675, 0.6273)	0.6643	0.7906	29.4893	0.4271
S6	0.4939 (0.3641, 0.6237)	0.6612	0.7876	29.4833	0.6683
S7	0.4970 (0.3671, 0.6269)	0.6640	0.7887	29.5248	0.9296
S8	0.4780 (0.3483, 0.6077)	0.6469	0.7772	29.4381	0.0100
S1-TTA	0.4947 (0.3649, 0.6245)	0.6620	0.7788	29.2889	0.7959
S2-TTA	0.5107 (0.3809, 0.6405)	0.6762	0.7943	29.4858	0.6633
S3-TTA	0.5110 (0.3812, 0.6408)	0.6764	0.7950	29.5209	0.4975
S4-TTA	0.5000 (0.3701, 0.6299)	0.6667	0.7926	29.5162	0.4975
S5-TTA	0.5031 (0.3732, 0.6330)	0.6694	0.7952	29.5535	0.4975
S6-TTA	0.5020 (0.3721, 0.6319)	0.6684	0.7920	29.5307	0.4271
S7-TTA	0.5083 (0.3785, 0.6381)	0.6740	0.7944	29.5845	0.4925
S8-TTA	0.4872 (0.3574, 0.6170)	0.6552	0.7888	29.5341	0.3878
S2, S3-TTA	0.5174 (0.3876, 0.6472)	0.6819	0.7997	29.6055	0.5779
S2, S3, S5-TTA	0.5182 (0.3884, 0.6480)	0.6827	0.8002	29.6076	0.5126
S2, S3, S5, S7-TTA	**0.5200 (0.3902, 0.6498)**	**0.6842**	0.8007	29.6174	0.4925
S2, S3, S5, S7, S6-TTA	**0.5200 (0.3902, 0.6498)**	**0.6842**	**0.8018**	**29.6408**	0.4874
S2, S3, S5, S7, S6, S4-TTA	0.5193 (0.3895, 0.6491)	0.6836	0.8009	29.6186	0.4925

## Data Availability

The data required to reproduce this study is publicly available and cited in the manuscript.

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
