# Peer review of "Assessing the Impact of Image Resolution on Deep Learning for TB Lesion Segmentation on Frontal Chest X-rays"

_diagnostics, 2023, doi:10.3390/diagnostics13040747_

Round 1

Reviewer 1 Report

I reviewed the study titled “Assessing the impact of image resolution on Deep Learning for TB lesion segmentation on frontal chest X-rays” in detail. In the current study, researchers investigated performance changes using an Inception-V3 UNet model using a variety of image resolutions with or without lung ROI cropping and aspect ratio adjustments. In the study, the researchers used the Shenzhen CXR dataset, which includes 326 normal patients and 336 TB patients. The points I was concerned about in the study were presented in articles.

The literature review on the subject should be expanded. The study's contribution to the literature and the novelty of the study should be presented at the end of the introduction section. The model used in the study should be included in the abstract section. By which method were the heatmap images presented in Figure 1 obtained? The environment in which the application results were obtained should be added to section 2.2 of the study. The results were obtained by using the inceptionv3 model in the study. There is not much innovation at this stage. For this reason, table 2 should be detailed. Limitations of the study should be addressed.

Author Response

I reviewed the study titled “Assessing the impact of image resolution on Deep Learning for TB lesion segmentation on frontal chest X-rays” in detail. In the current study, researchers investigated performance changes using an Inception-V3 UNet model using a variety of image resolutions with or without lung ROI cropping and aspect ratio adjustments. In the study, the researchers used the Shenzhen CXR dataset, which includes 326 normal patients and 336 TB patients. The points I was concerned about in the study were presented in articles.

Author response: We render our sincere thanks to the reviewer for the insightful comments and for encouraging the resubmission of our manuscript. To the best of our knowledge and belief, we have addressed the concerns of the reviewer in the revised manuscript.

Q1: The literature review on the subject should be expanded.

Author response: We agree with the reviewer's suggestions. We have created a new section entitled “Related literature and contributions of the study” to discuss the related literature, and list the contributions of our study. We also added recent publications related to the topic under discussion.  

Q2: The study's contribution to the literature and the novelty of the study should be presented at the end of the introduction section. The model used in the study should be included in the abstract section.

Author response: Thanks. The limitations of the current literature and the contributions of this study are discussed at the end of the related literature Section as follows:

Our review of the literature also revealed that identifying the optimal image resolution for the task under study remains an open avenue for research. Until the writing of this manuscript, we have not found any study that discussed the impact of image resolution on a CXR-based segmentation task, particularly for segmenting TB-consistent lesions. To close this gap in the literature, this work aims to study the impact of training a model on varying image resolutions with/without lung ROI cropping and aspect ratio adjustments to find the optimal resolution that improves fine-grained TB-consistent lesion segmentation. Further, this work proposes to improve performance at the optimal resolution through a combinatorial approach consisting of storing model snapshots, optimizing the test-time augmentation (TTA) methods, optimizing the segmentation threshold, and averaging the predictions of the model snapshots.

The model used in the study is included in the abstract Section as follows:

In this study, we investigated the performance variations using an Inception-V3 UNet model using various image resolutions with/without lung ROI cropping and aspect ratio adjustments, and (ii) identified the optimal image resolution through extensive empirical evaluations to improve TB-consistent lesion segmentation performance.

Q3: By which method were the heatmap images presented in Figure 1 obtained?

Author response: The binarized TB masks of men and women were resized to 256×256 pixel resolution to maintain uniformity in scale. Then, the masks were averaged, normalized to the range [0, 1], and displayed using the “jet” colormap.

Q4: The environment in which the application results were obtained should be added to section 2.2 of the study.

Author response: Thanks. We have included the details about the environment as follows:

The models were trained using Keras with Tensorflow backend (ver. 2.7) using a single NVIDIA GTX 1080 Ti GPU and CUDA dependencies.

287 CXRs and their associated TB masks were directly down-sampled using bi-cubic interpolation to the aforementioned resolutions. The OpenCV package (ver. 4.5.4) was used in this regard.

Q5: The results were obtained by using the inceptionv3 model in the study. There is not much innovation at this stage. For this reason, table 2 should be detailed.

Author response: Thanks for these insightful comments. The choice of model architecture depends on the characteristics of the data under study. As we discussed in Section 2.2, the Inception-V3 UNet model architecture was used since it delivered superior TB-consistent lesion segmentation performance in our previous study [12]. This study does not aim to propose a novel architecture, but to emphasize the need to study the data characteristics, while investigating the impact of varying image resolutions with/without ROI cropping as well as aspect ratio adjustments on the model’s performance.

Q6: Limitations of the study should be addressed.

Author response: Agreed. The limitations are discussed in the revised manuscript as follows:

We did not observe a considerable performance improvement with aspect ratio corrections. We were constrained by the UNet architecture [23] that requires that the length and width of the images/masks should be divisible by 32. This limitation did not allow us to make precise aspect ratio corrections. However, the study of literature [22] reveals that DL models trained on medical images are robust to changes in the aspect ratio.

Another limitation is that our experiments and conclusions are based on the Shenzhen CXR dataset where we observed that segmenting TB-consistent lesions using an UNet model trained on lung-cropped CXRs/masks delivers optimal performance at the 256×256 image resolution. These observations could vary across the datasets. We, therefore, emphasize that the characteristics of the data under study, the model performances at varying image resolutions with/without ROI cropping, and aspect ratio adjustments, should be discussed in all studies.

Due to GPU constraints, we were not able to train high-resolution models at larger batch sizes. However, with the advent of high-performance computing, this can be made feasible. High-resolution datasets might require newer model architecture and hardware advancements. Nevertheless, although the full potential of high-resolution datasets is not explored yet, it is indispensable to collect data at the highest resolution possible.  Additionally, irrespective of the image resolution, adding more experts to the annotation process may reduce the variation in the ground truth which we believe may improve segmentation performance.

Reviewer 2 Report

This paper proposed “Assessing the impact of image resolution on Deep Learning for TB lesion segmentation on frontal chest X-rays”. The approach discussed in this manuscript is interesting. To enhance the quality of the research, I recommend following corrections.

1-     In this area, a lot of work has been done, and a lot of important work is missing. Introduction section needs more investigation of some latest and relevant work.

2-     Literature review not fully explored and what is so far work has been carried out in this area not fully covered. I suggest the authors to add related work section and come up with clear flaws in state of art approaches and justification of their work. I suggest some article (Deep learning model integrating features and novel classifiers fusion for brain tumor segmentation, Brain tumor detection and multi-classification using advanced deep learning techniques).

3-     Explain your proposed and based algorithm separately with example that is easily understand for reader. Write a pseudocode of your proposed method and based method and clearly define your contribution.

4-     You have applied the DL model and the dataset is not sufficient and unknown, use some standard open-source dataset and compare your result with a well-known dataset that shows your work credibility.

5-     Even its not compulsory I suggest you upload your source code with manuscript.

6-     There are ambiguities and unclear meanings throughout the whole paper. Be more precise with your language.

Author Response

This paper proposed “Assessing the impact of image resolution on Deep Learning for TB lesion segmentation on frontal chest X-rays”. The approach discussed in this manuscript is interesting. To enhance the quality of the research, I recommend following corrections.

Author response: We render our sincere thanks to the reviewer for these valuable and insightful comments and for encouraging the resubmission of our manuscript. To the best of our knowledge and belief, we have addressed the queries of the reviewer in the revised manuscript.

Q1: In this area, a lot of work has been done, and a lot of important work is missing. Introduction section needs more investigation of some latest and relevant work.

Author response: Thanks for these critical comments. Per the reviewer’s suggestion, we have created a new section entitled “Related literature and contributions of the study”. We also added recent publications related to the topic under discussion as shown below: 

  1. Iqbal S, Ghani Khan MU, Saba T, Mehmood Z, Javaid N, Rehman A, Abbasi R. Deep learning model integrating features and novel classifiers fusion for brain tumor segmentation. Microsc Res Tech. 2019 Aug;82(8):1302-1315. doi: 10.1002/jemt.23281. Epub 2019 Apr 29. PMID: 31032544.
  2. Sadad T, Rehman A, Munir A, Saba T, Tariq U, Ayesha N, Abbasi R. Brain tumor detection and multi-classification using advanced deep learning techniques. Microsc Res Tech. 2021 Jun;84(6):1296-1308. doi: 10.1002/jemt.23688. Epub 2021 Jan 5. PMID: 33400339.
  3. Khan AR, Khan S, Harouni M, Abbasi R, Iqbal S, Mehmood Z. Brain tumor segmentation using K-means clustering and deep learning with synthetic data augmentation for classification. Microsc Res Tech. 2021 Jul;84(7):1389-1399. doi: 10.1002/jemt.23694. Epub 2021 Feb 1. PMID: 33524220.

Q2: Literature review not fully explored and what is so far work has been carried out in this area not fully covered. I suggest the authors to add related work section and come up with clear flaws in state of art approaches and justification of their work. I suggest some article (Deep learning model integrating features and novel classifiers fusion for brain tumor segmentation, Brain tumor detection and multi-classification using advanced deep learning techniques).

Author response: Thanks. We have added the suggested literature and other recent publications in deep learning-based computer vision as shown below:

  1. Iqbal S, Ghani Khan MU, Saba T, Mehmood Z, Javaid N, Rehman A, Abbasi R. Deep learning model integrating features and novel classifiers fusion for brain tumor segmentation. Microsc Res Tech. 2019 Aug;82(8):1302-1315. doi: 10.1002/jemt.23281. Epub 2019 Apr 29. PMID: 31032544.
  2. Sadad T, Rehman A, Munir A, Saba T, Tariq U, Ayesha N, Abbasi R. Brain tumor detection and multi-classification using advanced deep learning techniques. Microsc Res Tech. 2021 Jun;84(6):1296-1308. doi: 10.1002/jemt.23688. Epub 2021 Jan 5. PMID: 33400339.
  3. Khan AR, Khan S, Harouni M, Abbasi R, Iqbal S, Mehmood Z. Brain tumor segmentation using K-means clustering and deep learning with synthetic data augmentation for classification. Microsc Res Tech. 2021 Jul;84(7):1389-1399. doi: 10.1002/jemt.23694. Epub 2021 Feb 1. PMID: 33524220.

The limitations of the current literature and the contributions of this study are discussed at the end of the related literature Section as follows:

Our review of the literature also revealed that identifying the optimal image resolution for the task under study remains an open avenue for research. Until the writing of this manuscript, we have not found any study that discussed the impact of image resolution on a CXR-based segmentation task, particularly for segmenting TB-consistent lesions. To close this gap in the literature, this work aims to study the impact of training a model on varying image resolutions with/without lung ROI cropping and aspect ratio adjustments to find the optimal resolution that improves fine-grained TB-consistent lesion segmentation. Further, this work proposes to improve performance at the optimal resolution through a combinatorial approach consisting of storing model snapshots, optimizing the test-time augmentation (TTA) methods, optimizing the segmentation threshold, and averaging the predictions of the model snapshots.

Q3: Explain your proposed and based algorithm separately with example that is easily understand for reader. Write a pseudocode of your proposed method and based method and clearly define your contribution.

Author response: Thanks for this insightful comment. Per the reviewer’s suggestion, we have included the pseudocode in the revised manuscript.

Q4: You have applied the DL model and the dataset is not sufficient and unknown, use some standard open-source dataset and compare your result with a well-known dataset that shows your work credibility.

Author response: The Shenzhen chest X-ray dataset is one of the popular, publicly available datasets that contain Chest X-rays manifesting TB-consistent abnormalities. The publication that released this dataset has been widely cited in the literature (https://scholar.google.com/citations?view_op=view_citation&hl=en&user=aKoHJp0AAAAJ&citation_for_view=aKoHJp0AAAAJ:hFOr9nPyWt4C). To the best of our knowledge, this is the only dataset that contains expert-annotated lung regions and fine-grained annotation of TB-consistent lesions. We used this dataset because we used expert-annotated lung regions to delineate the lung ROI and the fine-grained TB lesion annotations to train our segmentation models.

As we discussed in Section 3.2, the Inception-V3 UNet model architecture was used since it delivered superior TB-consistent lesion segmentation performance in our previous study [12]. This study does not aim to propose a novel architecture, but to emphasize the need to study the data characteristics, while investigating the impact of varying image resolutions with/without ROI cropping as well as aspect ratio adjustments on the model’s performance.

Q5: Even its not compulsory I suggest you upload your source code with manuscript.

Author response: Thanks. The source codes will be made publicly available at https://github.com/sivaramakrishnan-rajaraman/Image_resolution_aspect_ratio upon manuscript acceptance.

Q6: There are ambiguities and unclear meanings throughout the whole paper. Be more precise with your language.

Author response: Thanks. The revised manuscript has been proofread by a native English speaker and corrected for typos and grammatical errors.  

Round 2

Reviewer 1 Report

.

Reviewer 2 Report

No more comments